# LEARNING TO SEARCH EFFICIENT DENSENET WITH LAYER-WISE PRUNING

## ABSTRACT

Deep neural networks have achieved outstanding performance in many real-world applications with the expense of huge computational resources. The DenseNet, one of the recently proposed neural network architecture, has achieved the state-of-the-art performance in many visual tasks. However, it has great redundancy due to the dense connections of the internal structure, which leads to high computational costs in training such dense networks. To address this issue, we design a reinforcement learning framework to search for efficient DenseNet architectures with layer-wise pruning (LWP) for different tasks, while retaining the original advantages of DenseNet, such as feature reuse, short paths, etc. In this framework, an agent evaluates the importance of each connection between any two block layers, and prunes the redundant connections. In addition, a novel reward-shaping trick is introduced to make DenseNet reach a better trade-off between accuracy and float point operations (FLOPs). Our experiments show that DenseNet with LWP is more compact and efficient than existing alternatives.

## 1 INTRODUCTION

Deep neural networks are increasingly used on mobile devices, where computational resources are quite limited(Chollet, 2017; Sandler et al., 2018; Zhang et al., 2017; Ma et al., 2018). Despite the success of deep neural networks, it is very difficult to make efficient or even real-time inference on low-end devices, due to the intensive computational costs of deep neural networks. Thus, the deep learning community has paid much attention to compressing and accelerating different types of deep neural networks(Gray et al., 2017).

Among recently proposed neural network architectures, DenseNet (Huang et al., 2017b) is one of the most dazzling structures which introduces direct connections between any two layers with the same feature-map size. It can scale naturally to hundreds of layers, while exhibiting no optimization difficulties. In addition, it achieved state-of-the-art results across several highly competitive datasets. However, recent extensions of Densenet with careful expert design, such as Multi-scale DenseNet(Huang et al., 2017a) and CondenseNet(Huang et al., 2018), have shown that there exists high redundancy in DenseNet. Our paper mainly focuses on how to compress and accelerate the DenseNet with less expert knowledge on network design.

A number of approaches have been proposed to compress deep networks. Generally, most approaches can be classified into four categories: parameter pruning and sharing, low-rank factorization, transferred/compact convolutional filters, and knowledge distillation(Gray et al., 2017). Unlike these approaches requiring intensive expert experience, automatic neural architecture design has shown its potential in discovering powerful neural network architectures. Neural architecture search (NAS) has been successfully applied to design model architectures for image classification and language models (Liu et al., 2018; Zoph & Le, 2016; Pham et al., 2018; Liu et al., 2017a; Brock et al., 2017).

However, none of these NAS methods are efficient for DenseNet due to the dense connectivity between layers. It is thus interesting and important to develop an adaptive strategy for searching an on-demand neural network structure for DenseNet such that it can satisfy both computational budget and inference accuracy requirement.

To this end, we propose a layer-wise pruning method for DenseNet based on reinforcement learning. Our scheme is that an agent learns to prune as many as possible weights and connections while maintaining good accuracy on validation dataset. As illustrated in Figure 1, our agent learns to output a sequence of actions and receives reward according to the generated network structure on validation datasets. Additionally, our agent automatically generates a curriculum of exploration, enabling effective pruning of neural networks.

Extensive experiments on several highly competitive datasets show that our method largely reduces the number of parameters as well as flops, while maintaining or slightly degrading the prediction performance, such that the corresponding network architecture can adaptively achieve a balance between inference accuracy and computational resources.

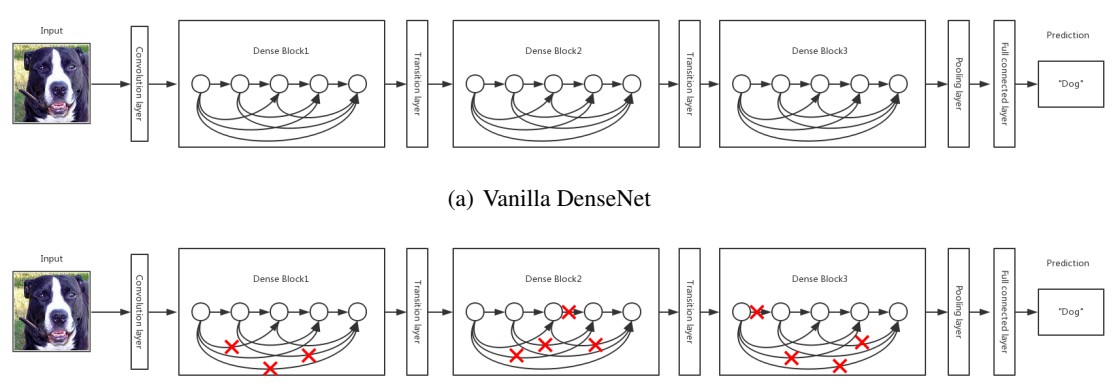

(a) Vanilla DenseNet

(b) DenseNet with layer-wise pruning

Figure 1: An illustration of layer-wise pruning method based on vanilla DenseNet. For one layer, not all connections are required and each layer has its unique connections. After being pruned some dense connections, the network can still predict correctly.

## 2 BACKGROUND

We first introduce reinforcement learning and policy gradient in Section 2.1, and DenseNet in Section 2.2, and finally neural architecture search in Section 2.3.

### 2.1 REINFORCEMENT LEARNING AND POLICY GRADIENT

Reinforcement learning considers the problem of finding an optimal policy for an agent that interacts with an uncertain environment and collects reward per action(Sutton et al., 1998). The goal of the agent is to maximize the long-term cumulative reward. Formally, this problem can be formulated as a Markov decision process over the environment states $s \in S$ and agent actions $a \in A$, under an unknown environmental dynamic defined by a transition probability $T(s'|s, a)$ and a reward signal $r(s, a)$ immediately following the action $a$ performed at state $s$. The agent's action $a$ is selected by a conditional probability distribution $\pi(a|s)$ called policy or actor. In policy gradient methods, we consider a set of candidate policies $\pi_\theta(a|s)$ parameterized by $\theta$ and obtain the optimal policy by maximizing the expected cumulative reward or return

$$J(\theta) = \mathbb{E}_{s\sim\rho_\pi, a\sim\pi(a|s)} \left[ r(s, a) \right],$$ (1)

where $\rho_\pi(s) = \sum_{t=1}^{\infty} \gamma^{t-1} \Pr(s_t = s)$ is the normalized discounted state visit distribution with a discount factor $\gamma \in [0, 1)$. To simplify the notation, we denote $\mathbb{E}_{s \sim \rho_\pi, a \sim \pi(a|s)}[\cdot]$ by simply $\mathbb{E}_\pi[\cdot]$ in the rest of paper.

According to the policy gradient theorem(Sutton et al., 1998), the gradient of $J(\theta)$ can be written as

$$\nabla_\theta J(\theta) = \mathbb{E}_\pi \left[ \nabla_\theta \log \pi(a|s) Q^\pi(s, a) \right], \tag{2}$$

where $Q^\pi(s, a) = \mathbb{E}_\pi \left[ \sum_{t=1}^{\infty} \gamma^{t-1} r(s_t, a_t) | s_1 = s, a_1 = a \right]$ denotes the expected return under policy $\pi$ after taking an action $a$ in state $s$, which is also called critic.

Since the expectation in Eq (3) is over action, it is helpful to estimate a value function $V(s)$ and subtract it from $Q(s, a)$ to reduce variance while keeping unbiased.

$$\nabla_\theta J(\theta) = \mathbb{E}_\pi \left[ \nabla_\theta \log \pi(a|s) \left( Q^\pi(s, a) - V(s) \right) \right]. \tag{3}$$

The most straightforward way is to simulate the environment with the current policy $\pi$ to obtain a trajectory $\{(s_t, a_t, r_t)\}_{t=1}^{n}$ and estimate $\nabla_\theta J(\theta)$ using the Monte Carlo estimation:

$$\hat{\nabla}_\theta J(\theta) = \frac{1}{n} \sum_{t=1}^{n} \gamma^{t-1} \nabla_\theta \log \pi(a_t|s_t) \left( \hat{Q}^\pi(s_t, a_t) - \hat{V}(s_t) \right), \tag{4}$$

where $\hat{Q}^\pi(s_t, a_t)$ is an empirical estimate of $Q^\pi(s_t, a_t)$, e.g., $\hat{Q}^\pi(s_t, a_t) = \sum_{j \geq t} \gamma^{j-t} r_j$, and $\hat{V}(s_t)$ is an empirical estimate of $V(s)$.

## 2.2 DENSENET

Densely connected networks(Huang et al., 2017b) consist of multiple dense blocks, each of which also consists of multiple layers. Each layer produces $k$ features maps, where $k$ is referred to the growth rate of the network. The distinguishing property of DenseNets is that the input of each layer is a concatenation of all feature maps generated by all preceding layers within the same dense block.

Inside every dense block, the first transformation is a composition of batch normalization(BN) and rectified linear units(RELU), followed by the first convolutional layer in the sequence which reduces the number of channels to save computational cost by using the $1 \times 1$ filters. The output is then followed by another BN-ReLU combination transformation and is then reduced to the final $k$ output features through a $3 \times 3$ convolution.

## 2.3 NEURAL ARCHITECTURE SEARCH

Neural Architecture Search(NAS) is a method for automated design of neural network structures, with the aid of either evolutionary algorithms(Xie & Yuille, 2017; Real et al., 2017) or reinforcement learning (Baker et al., 2016; Cai et al., 2018; Zhong et al., 2017; Zoph & Le, 2016; Zoph et al., 2017). When using reinforcement learning, the agent performs a sequence of actions, which specifies a network structure; this network is then trained and its corresponding validation performance is returned as the reward to update the agent.

## 3 METHOD

We analyze the dense connections of DenseNet in Section 3.1, then we model the layer-wise pruning as a Markov decision process (MDP)(Puterman, 2014) and design a Long-short term memory( LSTM)(Hochreiter & Schmidhuber, 1997) controller to generate inference paths in Section 3.2. The interaction between the agent (i.e., the LSTM controller) and the environment (i.e., the DenseNet) is described in Figure 2. The reward shaping technique in our method is introduced in Section 3.3. Finally, we show the complete training process of LWP in Section 3.4.

### 3.1 Pretrained Dense Convolutional Networks

Vanilla DenseNet consists of four parts: the first convolution layer, multiple dense blocks, transition layers and finally the fully-connected layer. The first convolution layer is only for feature extraction from raw data. As for the multiple dense blocks, each dense block consists of multiple layers. The transition layers are used as down-sampling layers to change the size of feature maps and the last full-connected layer is used for image classification. Obviously, the dense connections are mainly reflected on the dense blocks. Therefore, we study the connection policy for dense layers in this paper.

### 3.2 Generate Inference Paths with an LSTM Controller

Suppose the DenseNet has $L$ layers, the controller needs to make $K$ (equal to the number of layers in dense blocks) decisions. For layer $i$, we specify the number of previous layers to be connected in the range between 0 and $n_i$ ($n_i = i$). All possible connections among the DenseNet constitute the action space of the agent. However, the time complexity of traversing the action space is $\mathcal{O}(\prod_{i=1}^{K} 2^{n_i})$, which is NP-hard and unacceptable for DenseNet(Huang et al., 2017b). Fortunately, reinforcement learning is good at solving sequential decision optimization problems and we model the network pruning as a Markov Decision Process(MDP). Since these hierarchical connections have time-series dependencies, it is natural to train LSTM as the controller to simply solve the above-mentioned issue.

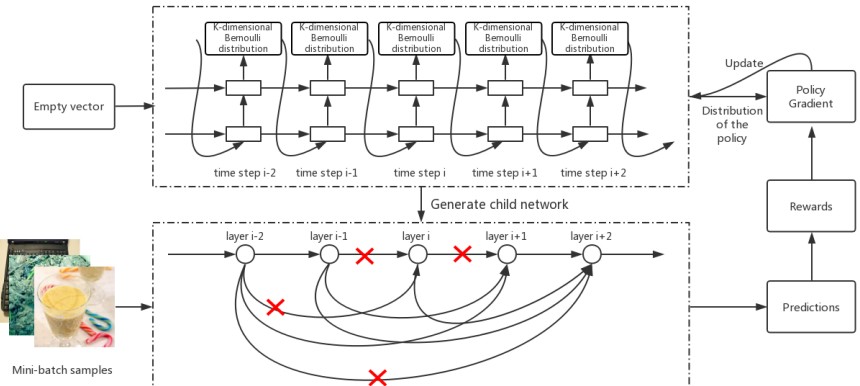

Figure 2: Illustration of our proposed framework. In each iteration, the output of the $i$-th time step makes keeping or dropping decisions for the $i$-th layer. All outputs of the LSTM controller generate a child network by sampling from $K \times K$-dimensional Bernoulli distribution. Then, the child network forwards propagation with mini-batch samples and the reward function can be evaluated with the predictions and FLOPs. The controller is optimized with policy gradient.

At the first time step, the LSTM controller receives an empty embedding vector as the input that is regarded as the fixed state $\mathbf{s}$ of the agent, and the output of the previous time step is the input for the next time step. Each output neuron in the LSTM is equipped with $\delta(x) = \frac{1}{1+e^{-x}}$, so that the output $\mathbf{o_i}$ defines a policy $p_{i,\mathbf{a_i}}$ of keeping or dropping connections between the current layer and its previous layers as an $n_i$-dimensional Bernoulli distribution:

$$\mathbf{o_i} = f(\mathbf{s}; \theta_{\mathbf{c}}), \qquad p_{i,\mathbf{a_i}} = \prod_{j=1}^{n_i} o_{ij}^{a_{ij}} (1 - o_{ij})^{(1-a_{ij})}, \qquad (5)$$

where $f$ denotes the controller parameterized with $\theta_{\mathbf{c}}$. The $j$-th entry of the output vector $\mathbf{o_i}$, denoted by $o_{ij} \in [0, 1]$, represents the likelihood probability of the corresponding connection between the $i$-th layer and

the $j$-th layer being kept. The action $\mathbf{a_i} \in \{0,1\}^{n_i}$ is sampled from Bernoulli($\mathbf{o_i}$). $a_{ij} = 1$ means keeping the connection, otherwise dropping it. There are total $n_i$ connections for the $i$-th layer, but the output dimension of LSTM at each time step is $K$. To unify the action space dimension and LSTM output dimension, we set both to $K$ and the output of each time step take a $mask \in \{0,1\}^K$ operation, where the mask numbers from $1$-$th$ to $n_i$-$th$ element are 1 and others are 0. Finally, the probability distribution of the whole neural network architecture is formed as:

$$\pi(\mathbf{a}_{1:K}|s;\theta_\mathbf{c}) = \prod_{i=1}^{K} p_{i,\mathbf{a}_i} \tag{6}$$

### 3.3 REWARD SHAPING

Reward shaping is introduced to help the controller make progress to an optimal solution. The reward function is designed for each sample and not only considers the prediction correct or not, but also encourages less computation:

$$R(a) = \begin{cases} 1 - \eta^\alpha & \text{if predict correctly} \\ -\gamma & \text{otherwise.} \end{cases} \tag{7}$$

where $\eta = \frac{SUBFLOPs}{FLOPs}$ measures the percentage of float operations utilized. SUBFLOPs, FLOPs represent the float point operations of the child network and vanilla DenseNet, respectively. In order to maximize the reward, the prediction needs to be correct and SUBFLOPs should be reduced as much as possible. The trade-off between performance and complexity is mainly controlled by $\alpha$ and $\gamma$ and more details will be discussed in the Section 7.3 of the appendix.

### 3.4 TRAINING WITH ADVANTAGE ACTOR-CRITIC

After obtaining the feedback from the child network, we modify the Eq (1) as the following expected reward:

$$J(\theta_c) = \mathbb{E}_{a \sim \pi_{\theta_c}}[r(s,a)] \tag{8}$$

To maximize Eq (8) and accelerate policy gradient training over $\theta_\mathbf{c}$, we utilize the advantage actor-critic(A2C) with an estimation of state value function $V(s;\theta_v)$ to derive the gradients of $J(\theta_c)$ as:

$$\nabla_{\theta_c} J(\theta_c) = \sum_a (r(s,a) - V(s;\theta_v)) \pi(a|s,\theta_c) \nabla_{\theta_c} \log \pi(a|s,\theta_c) \tag{9}$$

The Eq (9) can be approximated by using the Monte Carlo sampling method:

$$\nabla_{\theta_c} J(\theta_c) = \frac{1}{n} \sum_{t=1}^{n} \left( r^{(t)}(s,a) - V(s;\theta_v) \right) \nabla_{\theta_c} \log \pi(a|s,\theta_c) \tag{10}$$

where $n$ is the batch size. The mini-batch samples share the same child network and perform forward propagation in parallel. Therefore, they have the same policy distribution $\pi(a|s,\theta_c)$ but different $r(s,a)$. We further improve exploration to prevent the policy from converging to suboptimal deterministic policy by adding the entropy of the policy $\pi(a|s,\theta_c)$,$H(\pi(a|s,\theta_c))$ to the objective function. The gradient of the full objective function takes the form:

$$\nabla_{\theta_c} J(\theta_c) = \frac{1}{n} \sum_{t=1}^{n} \left[ \left( r^{(t)}(s,a) - V(s,\theta_v) \right) \nabla_{\theta_c} \log \pi(a|s,\theta_c) + \beta \nabla_{\theta_c} H(\pi(a|s,\theta_c)) \right] \tag{11}$$

As for the value network, we define the loss function as $L_v$ and utilize gradient descent methods to update $\theta_v$:

$$L_v = \frac{1}{n} \sum_{t=1}^{n} \left( r^{(t)}(s,a) - V(s;\theta_v) \right)^2, \qquad \nabla_{\theta_v} L_v = \frac{2}{n} \sum_{t=1}^{n} \left( r^{(t)}(s,a) - V(s;\theta_v) \right) \frac{\partial V(s;\theta_v)}{\partial \theta_v} \tag{12}$$

The entire training procedure is divided into three stages: curriculum learning, joint training and training from scratch. Algorithm 1 shows the complete recipe for layer-wise pruning.

**Curriculum learning.** It is easy to note that the search space scales exponentially with the block layers of DenseNet and there are total $\prod_{i=1}^{K} 2^{n_i}$ keeping/dropping configurations. We use curriculum learning(Bengio, 2013) like BlockDrop(Wu et al., 2018) to solve the problem that policy gradient is sensitive to initialization. For epoch $t$ ($1 \leq t < K$), the LSTM controller only learns the policy of the last $t$ layers and keeps the policy of the remaining $K - t$ layers consistent with the vanilla DenseNet. As $t \geq K$, all block layers are involved in the decision making process.

**Joint training.** The previous stage just updates parameters $\theta_c$ and $\theta_v$. The controller learns to identify connections between two block layers to be kept or dropped. However, it prevents the agent from learning the optimal architecture. Jointly training the DenseNet and controller can be employed as the next stage so that the controller guides the gradients of $\theta_v$ to the direction of dropping more connections.

**Training from scratch.** After joint training, several child networks can be sampled from the policy distribution $\pi(a|s, \theta_c)$ and we select the child network with the highest reward to train from scratch, and thus better experimental results have been produced.

We summarize the entire process in Algorithm 1.

## 4 RELATED WORK

Huang et al. (2018) proposed group convolution to remove connections between layers in DenseNet for which this feature reuse is superfluous; Huang et al. (2017a) also suggested progressively update prediction for every test sample to unevenly adapt the amount of computational resource at inference time. The most related work is BlockDrop (Wu et al., 2018), which used reinforcement learning to prune weight dynamically at inference time but can only be applied to ResNet or its variants. In contrast, our approach is based on DenseNet, aiming to find efficient network structure based the densely connected features of DenseNet.

## 5 EXPERIMENT

We evaluate the LWP method on three benchmarks: CIFAR-10, CIFAR-100 (Krizhevsky & Hinton, 2009) and ImageNet 2012 (Deng et al., 2009) and these three datasets are used for image classification. Details of experiments and hyperparameters setting in Appendix 7.3.

### 5.1 RESULTS ON CIFAR

**Pretrained DenseNet.** For CIFAR datasets, DenseNet-40-12 and DenseNet-100-12 are selected as the backbone CNN. During the training time, the backbone CNN needs to make predictions with dynamic computation paths. In order to make the backbone CNN adjust to our algorithm strategy, we reproduced the DenseNet-40-12 and DenseNet-100-12 on CIFAR based on Pytorch (Paszke et al., 2017) and the results are shown in Table 1.

**Comparisons and analysis.** The results on CIFAR are reported in Table 1. For CIFAR-10 dataset and the vanilla DenseNet-40-12, our method has reduced the amounts of FLOPs, parameters by nearly $81.4\%$, $78.2\%$, respectively and the test error only increase $1.58\%$. The exponential power $\alpha$ and penalty $\gamma$ can be tuned to improve the performance. In this experiment, we just modify hyperparameter $\alpha$ from 2 to 3 so that the model

complexity($105M$ vs $173M$ FLOPs) is increased while test error rate is reduced to $6.00\%$.The same law can be observed on the DenseNet-100-12 with LWP. Our algorithm also has advantages on Condensenet (Huang et al., 2018) which needs more expert knowledge and NAS (Zoph & Le, 2016) which takes much search time complexity and needs more parameters but gets higher test error.

We can also observe the results on CIFAR-100 from the Table 1 that the amounts of FLOPs in DenseNet with LWP are just nearly $46.5\%$, $66.3\%$ of the DenseNet-40-12 and DenseNet-100-12. The compression rates are worse than that for CIFAR-10. This may be caused by the complexity of the CIFAR-100 classification task. The more hard task, the more computation is needed.

| Model | FLOPs | Params | CIFAR-10 | CIFAR-100 |
|---|---|---|---|---|
| DenseNet-40-12 (Huang et al., 2017b)(our impl.) | 566M | 1.10M | 5.24 | 25.09 |
| DenseNet-100-12 (Huang et al., 2017b)(our impl.) | 3.63G | 7.19M | 4.34 | 20.88 |
| VGG-16-Pruned (Li et al., 2016) | 206M | 5.40M | 6.60 | 25.28 |
| VGG-19-pruned (Liu et al., 2017b) | 195M | 2.30M | 6.20 | - |
| VGG-19-pruned (Liu et al., 2017b) | 250M | 5.00M | - | 26.52 |
| ResNet-110-pruned (Li et al., 2016) | 213M | 1.68M | 6.45 | - |
| DenseNet-40-pruned (Liu et al., 2017b) | 190M | 0.66M | 5.19 | 25.28 |
| CondenseNet$^{light}$-94 (Huang et al., 2018) | 122M | 0.33M | 5.00 | 24.08 |
| CondenseNet-86 (Huang et al., 2018) | 65M | 0.52M | 5.00 | 23.64 |
| NAS v2 predicting strides (Zoph & Le, 2016) | - | 2.5M | 6.01 | - |
| DenseNet-40-12-LWP ($\alpha = 2, \gamma = -0.5$) | 105M | 0.24M | 6.82 | - |
| DenseNet-40-12-LWP ($\alpha = 2, \gamma = -0.5$) | 263M | 0.66M | - | 26.99 |
| DenseNet-40-12-LWP ($\alpha = 3, \gamma = -0.5$) | 173M | 0.40M | 6.00 | - |
| DenseNet-100-12-LWP ($\alpha = 2, \gamma = -0.5$) | 716M | 1.43M | 5.12 | - |
| DenseNet-100-12-LWP ($\alpha = 2, \gamma = -0.5$) | 2.42G | 5.15M | - | 21.14 |

Table 1: Results on CIFAR. DenseNet-40-12 and DenseNet-100-12 are selected as the backbone CNN on CIFAR dataset and our algorithm is applied to the two models. The FLOPs, parameters and test error of the DenseNet with LWP are compered with the vanilla DenseNet and the neural network architecture with other pruned methods.

## 5.2 RESULTS ON IMAGENET

**Pretrained DenseNet.**  We compress the DenseNet-121-32 which has four dense blocks([6, 12, 24, 16]) on ImageNet. The growth rate of DenseNet-121-32 is 32 and this neural network architecture is equipped with bottleneck layers and compression ratio fixed at $0.5$ that are designed to improve the model compactness. In the following section, we prove that the model can be further compressed. This model is initialized by loading the checkpoint file of pretrained model from Pytorch.

**Make comparisons and analysis.**  Although the bottleneck layer and compression ratio are introduced in DenseNet-121-32, the result shows that there is still much redundancy. As observed from Table 2, we can still reduce $54.7\%$ FLOPs and $35.2\%$ parameters of the vanilla DenseNet-121-32 with $1.84\%$ top-1 and $1.28\%$ top-5 test error increasing.

| Model | FLOPs | Params | Top-1 | Top-5 |
|---|---|---|---|---|
| DenseNet-121-32-BC (Huang et al., 2017b) | 5.67G | 7.98M | 25.35 | 7.83 |
| DenseNet-121-32-BC-LWP | 2.57G | 5.17M | 27.19 | 9.11 |

Table 2: Results on ImageNet. DenseNet-121-32 is selected as the backbone CNN on ImageNet. It can be further compressed even if its parameters are already quite efficient.

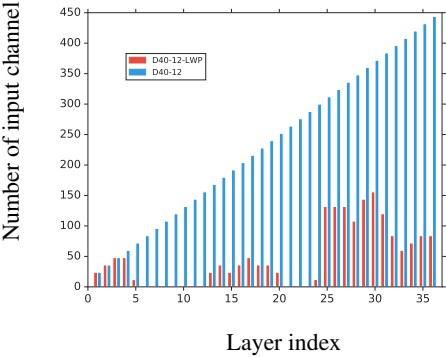 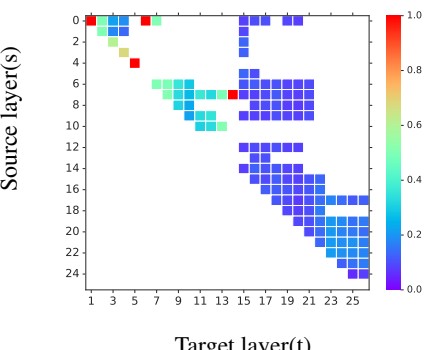

Figure 3: Quantitative results on DenseNet-40-12 wth LWP. $Left$: the number of input channel in vanilla DenseNet-40-12 and the learned child network. $Right$: the connection dependency between any two layers is represented as the average absolute wights of convolution layer.

## 5.3 QUANTITATIVE RESULTS

In this section, we argue that our proposed methods can learn more compact neural network architecture by analyzing the number of input channel in DenseNet layer and the connection dependency between a convolution layer with its preceding layers.

In Figure 3 $left$, the red bar represent the number of input channel in DenseNet-40-12-LWP (D40-12-LWP) and the blue bar represent the number of input channel in vanilla DenseNet. We can observe that the number of input channels grows linearly with the layer index because of the concatenation operation and D40-12-LWP has layer-wise input channels identified by the controller automatically. The input channel is $0$ means this layer is dropped so that the block layers is reduced from 36 to 26. The number of connections between a layer with its preceding layers can be obtained from the right panel of Figure 3. In Figure 3 $right$, the $x$, $y$ axis define the target layer $t$ and source layer $s$. The small square at position $(s, t)$ represents the connection dependency of target layer $t$ on source layer $s$. The pixel value of position $(s, t)$ is evaluated with the average absolute filter weights of convolution layers in D40-12-LWP. One small square means one connection and the number of small squares in the vertical direction indicates the number of connections to target layer $t$.

As reported by the paper DenseNet(Huang et al., 2017b), there are redundant connections because of the low kernel weights on average between some layers. The right panel of Figure 3 obviously shows that the values of these small square connecting the same target layer $t$ are almost equal which means the layer $t$ almost has the same dependency on different preceding layers. Naturally, we can prove that the child network learned from vanilla DenseNet is quite compact and efficient.

## 6 CONCLUSION

We propose an algorithm strategy to search efficient child network of DenseNet with reinforcement learning agent. The LSTM is used as the controller to layer-wise prune the redundancy connections. The whole process is divided into three stages: curriculum learning, joint training and training from scratch. The extensive experiments based on CIFAR and ImageNet show the effectiveness of our method. Analyzing the child network and the filter parameters in every convolution layer prove that our proposed method can learn to search compact and efficient neural network architecture.

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

# 7  APPENDIX

## 7.1  DATASETS AND EVALUATION METRICS

CIFAR-10 and CIFAR-100 consists of 10 and 100 classes images with $32 \times 32$ RGB pixels. Both datasets contain $60,000$ images, of which $50,000$ images for training sets and $10,000$ images for test sets. We use a standard data pre-processing and augmentation techniques and the complete procedure is: normalize the data by using the channel means and standard deviations, centrally pad the training images with size 4, randomly crop to restore $32 \times 32$ images and randomly flip with probability 0.5 horizontally. The evaluation metric in CIFAR is the prediction accuracy.

There are total $1.33$ million colored images with $1000$ visual classes in ImageNet, $1.28$ million for training images and $50k$ for validation images. We also adopt the data-augmentation scheme for pre-processing, ie: resize the images to $256 \times 256$, normalize the images using channel means and standard derivations, randomly crop to $224 \times 224$ and flip horizontally at training time but apply a center crop with size $224 \times 224$ at test time. The performance in ImageNet is measured by both top-1 and top-5 prediction accuracy.

## 7.2  TRAINING CONFIGURATIONS

**Training configurations for CIFAR.**  Based on the pretrained DenseNet, the LSTM controller is trained with batch size 128 for 1000 epochs during the curriculum learning procedure and ADAM optimizer without weight decay is adopt. The learning rate starts from $10^{-3}$ and it is lowered by 10 times at epoch 500 and 750. For the joint training, we fix the learning rate at $10^{-4}$ and finetune the model for 1000 epochs. Then we select the optimal child network with highest reward to train from scratch. In the last stage, the SGD optimizer with a weight decay of $10^{-4}$ and a Nesterov momentum of 0.9 without dampening is adopt. We train the optimal child network with mini-batch size 64 and a cosine shape learning rate from 0.1 to 0 for 300 epochs.

**Training configurations for ImageNet.**  For curriculum learning and joint training, we set the epochs 90, 50 respectively and batch size 1024. In curriculum learning procedure, the learning rate is set to $1e^{-3}$ and is lowered 10 times in epoch 45 and 75. The learning rate is fixed at $1e^{-4}$ for joint training procedure. We use the same optimizer parameters as CIFAR experiments. At last, the learned optimal child network is optimized like DenseNet(Huang et al., 2017b).

## 7.3  HYPERPARAMETERS SEARCH

We use reward shaping technique in our model and the detailed reward formulation is defined in Eq (7). The trade-off between the model performance and complexity can be controlled by adjusting different reward functions. As shown in Eq (7), the reward function mainly depends on the exponential power $\alpha$ of FLOPs ratio and the penalty $-\gamma$. We mainly analyzed and explored these two factors of the child network (D40-12-LWP) based on DenseNet-40-12 (D40-12) and CIFAR-10 dataset in the following section.

**Exponential power.**  Given one policy, pass a image to the child network and we hope to get higher reward if the prediction is correct. The lower the FLOPs of the child network, the larger the reward value if fix $\alpha$. On the contrary, in order to get the same reward value, the exponential power $\alpha$ is bigger and the model complexity is larger. As shown in Figure 4 (a) (b), with setting exponential power $\alpha = 1/3, 1/2, 1, 2, 3$ and fixing $\gamma = 0.5$, the classification accuracy and FLOPs of the child network also increase gradually.

**Penalty.**  Considering the incorrect prediction, the penalty $-\gamma$ is given as the feedback. The bigger $\gamma$ means the controller emphasis on the model performance rather than the reduced model complexity. $\gamma$ is set to

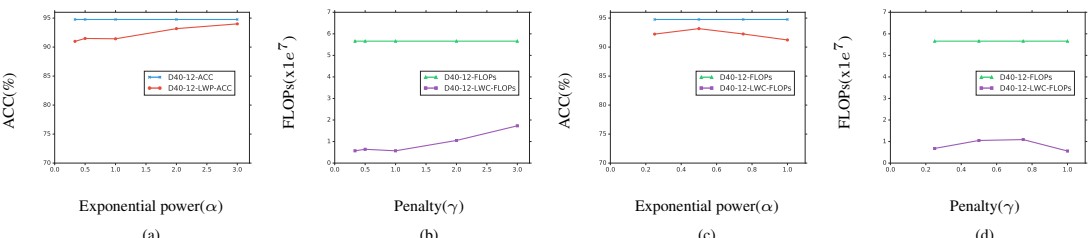

Figure 4: Hyperparameters search on CIFAR-10 to explore the trade off between FLOPs and accuracy by changing the exponential power of FLOPs ratio and penalty $-\gamma$.

$0.25, 0.5, 0.75, 1$ and exponential power $\alpha$ is fixed at 2, respectively. The results is shown in Figure 4 (c) (d) and we can observe that both curves are increased first and then decreased.

### 7.4 ALGORITHM

---
**Algorithm 1** The pseudo-code for layer-wise pruning.

---
**Input:**   Training dataset $\mathcal{D}_t$; Validation dataset $\mathcal{D}_v$; Pretrained DenseNet.
   Initialize the parameters $\theta_c$ of the LSTM controller and $\theta_v$ of the value network randomly.
   Set epochs for curriculum learning, joint training and training from scratch to $M^{cl}$, $M^{jt}$ and $M^{fs}$ respectively and sample $Z$ child networks.
**Output:**   The optimal child network
1: *//Curriculum learning*
2: **for** $t = 1$ to $M^{cl}$ **do**
3:     $\mathbf{o} = f(\mathbf{s}; \theta_c)$
4:     **if** $t < K - t$ **then**
5:         **for** $i = 1$ to $K - t$ **do**
6:             $\mathbf{o}[i, 0 : i] = 1$
7:             $\mathbf{o}[i, i :] = 0$
8:         **end for**
9:     **end if**
10:     Sample $\mathbf{a}$ from $Bernoulli(\mathbf{o})$
11:     DenseNet with policy makes predictions on the training dataset $\mathcal{D}_t$
12:     Calculate feedback $R(\mathbf{a})$ with Eq (7)
13:     Update parameters $\theta_c$ and $\theta_v$ with Eq (11) and Eq (12) respectively
14: **end for**
15: *//Joint training*
16: **for** $t = 1$ to $M^{jt}$ **do**
17:     Simultaneously train DenseNet and the controller
18: **end for**
19: **for** $t = 1$ to $Z$ **do**
20:     Sample a child network from $\pi(\mathbf{a}|\mathbf{s}, \theta_c)$
21:     Execute the child network on the validation dataset $\mathcal{D}_v$
22:     Obtain feedback $R^{(t)}(\mathbf{a})$ with Eq (7)
23: **end for**
24: Select the child network $\mathcal{N}$ with highest reward
25: *//Training from scratch*
26: **for** $t = 1$ to $M^{fs}$ **do**
27:     Train the child network $\mathcal{N}$ from scratch
28: **end for**
29: **return** The optimal child network $\mathcal{N}$

---

