# OpenReview forum: "Learning to Search Efficient DenseNet with Layer-wise Pruning"
_ICLR.cc/2019/Conference_

### Official Review · AnonReviewer2 · 2018-11-01
**The evaluation could be improved.**

**Rating:** 4
**Confidence:** 4

**Review:**

The paper introduces RL based approach to prune layers in a DenseNet. This work extends BlockDrop to DenseNet architecture making the controller independent form the input image. The approach is evaluated on CIFAR10 and CIFAR100 datasets as well as on ImageNet showing promising results.

In order to improve the paper, the authors could take into consideration the following points:

1. Given the similarity of the approach with BlockDrop, I would suggest to discuss it in the introduction section clearly stating the similarities and the differences with the proposed approach.
2. BlockDrop seems to introduce a general framework of policy network to prune neural networks. However, the authors claim that BlockDrop "can only be applied to ResNets or its variants". Could the authors comment on this?
3. In the abstract, the authors claim: "Our experiments show that DenseNet with LWP is more compact and efficient than existing alternatives". It is hard to asses if the statement is correct given the evidence presented in the experimental section. It is not clear if the method is more efficient and compact than others, e. g.  CondenseNet.
4. In the experimental section, addressing the following questions would make the section stronger: What is more important FLOPs or number of parameters? What is the accuracy drop we should allow to pay for reduction in number of parameters or FLOPs?
5. For the evaluation, I would suggest to show that the learned policy is better than a random one: e. g. not using the controller to define policy (in line 20 of the algorithm) and using a random random policy instead.
6. In Table 1, some entries for DenseNet LWP are missing. Is the network converging for this setups?
7. \sigma is not explained in section 3.3. What is the intuition behind this hyper parameter?
8. I'd suggest moving related work section to background section and expanding it a bit.
9. In the introduction: "... it achieved state-of-the-art results across several highly competitive datasets". Please add citations accordingly.

Additional comments:
1. It might be interesting to compare the method introduced in the paper to a scenario where the controller is conditioned on an input image and adaptively selects the connections/layers in DenseNet at inference time.
2. It might be interesting to report the number of connections in Table 1 for all the models.

Overall, I liked the ideas presented in the paper. However, I think that the high degree of overlap with BlockDrop should be addressed by clearly stating the differences in the introduction section. Moreover, I encourage the authors to include missing results in Table 1 and run a comparison to random policy. In the current version of the manuscript, it is hard to compare among different methods, thus, finding a metric or a visualization that would clearly outline the "efficiency and compactness" of the method would make the paper stronger.

---

### Official Review · AnonReviewer1 · 2018-11-04
**RL based method for pruning a pre-trained network**

**Rating:** 4
**Confidence:** 4

**Review:**

This paper proposes a layer-based pruning method based on reinforment learning for pre-train networks.

There are several major issues for my rating:

- Lack of perspective. I do not understand where this paper sits compared to other compression methods. If this is about RL great, if this is about compression, there is a lack of related work and proper comparisons to existing methods (at least concenptual)
- Claims about the benefits of not needed expertise are not clear to me as, from the results, seems like expertise is needed to set the hyperparameters.

- experiments are not convincing. I would like to see something about computational costs. Current methods aim at minimizing training / finetuning costs while maintaining the accuracy. How does this stands in that regard? How much time is needed to prune one of these models? How many resources?

- Would it be possible to add this process into a training from scratch method?

- how would this compare to training methods that integrate compression strategies?
- Table 1 shows incomplete results, why? Also, there is a big gap between accuracy/number of parameters trade-of between this method and other presented in that table. Why?

---

### Official Review · AnonReviewer3 · 2018-11-06
**Straightforward Idea with Limited Contribution**

**Rating:** 4
**Confidence:** 5

**Review:**

This paper proposes to apply Neural Architecture Search (NAS) for connectivity pruning to improve the parameter efficiency of DenseNet. The idea is straightforward and the paper is well organized and easy to follow.

My major concern is the limited contribution. Applying deep reinforcement learning (DRL) and following the AutoML framework for architecture/parameter pruning has been extensively investigated during the past two years. For instance, this work has a similar motivation and design "AMC: AutoML for Model Compression and Acceleration on Mobile Devices."

The experimental results also show a limited efficiency improvement according to Table 1. Although this is a debatable drawback compared with the novelty/contribution concern, it worth to reconsider the motivation of the proposed method given the fact that the AutoML framework is extremely expensive due to the DRL design.

---

### Meta-Review · Area_Chair1 · 2018-12-12
**Limited contribution**

**Confidence:** 5
**Recommendation:** Reject

**Metareview:**

The paper proposes to apply Neural Architecture Search for pruning DenseNet.

The reviewers and AC note the potential weaknesses of the paper in various aspects, and decided that the authors need more works to publish.